# Collagen Nanoparticles in Drug Delivery Systems and Tissue Engineering

**Ashni Arun** [1,†], **Pratyusha Malrautu** [1,†], **Anindita Laha** [1,*], **Hongrong Luo** [2,*] and **Seeram Ramakrishna** [3]

1   Department of Chemical Engineering, Manipal Institute of Technology, Manipal Academy of Higher Education, Manipal 576104, India; ashniarun2018@gmail.com (A.A.); pratyusha.malrautu99@gmail.com (P.M.)
2   Engineering Research Center in Biomaterials, Sichuan University, Chengdu 610064, China
3   Center of Nanofibers and Nanotechnology, National University of Singapore, Singapore 117581, Singapore; seeram@nus.edu.sg
*   Correspondence: anindita.laha19@gmail.com (A.L.); hluo@scu.edu.cn (H.L.)
†   Both authors contributed equally.

**Abstract:** The versatile natural polymer, collagen, has gained vast attention in biomedicine. Due to its biocompatibility, biodegradability, weak antigenicity, biomimetics and well-known safety profile, it is widely used as a drug, protein and gene carrier, and as a scaffold matrix in tissue engineering. Nanoparticles develop favorable chemical and physical properties such as increased drug half-life, improved hydrophobic drug solubility and controlled and targeted drug release. Their reduced toxicity, controllable characteristics of scaffolds and stimuli-responsive behavior make them suitable in regenerative medicine and tissue engineering. Collagen associates and absorbs nanoparticles leading to significant impacts on their biological functioning in any biofluid. This review will discuss collagen nanoparticle preparation methods and their applications and developments in drug delivery systems and tissue engineering.

**Keywords:** drug delivery systems (DDS); tissue engineering (TE); polymers; collagen; nanoparticle (NP)

## 1. Introduction

Collagen is an abundant protein found in the human body existing in bones, tendons, muscles and skin [1]. Properties such as tensile strength, biodegradability, notable stretchability, absorption in-vivo capacity, biocompatibility, biomimetic nature, weak antigenicity, and remarkable safety profile make it a significant element in applications of drug delivery systems (DDS) and tissue engineering (TE) [2]. Collagen has good in vivo absorption, synergism with bioactive materials, haemostatic characteristics, and low immunogenicity [3,4]. To overcome limitations such as enzymatic degradation, weak mechanical strength and low thermal stability, methods such as crosslinking, grafting polymerization, blending and covalent conjugation are adopted, resulting in biomaterials of discrete physical forms such as shields, films, sponges, hydrogels, microspheres, sheets, coatings, liposomes, disks, nanofibers, tablets, pellets and nanoparticles (NPs) [5,6].

Collagen acts as an efficient carrier for the delivery of various agents such as genes, drugs, proteins and growth factors. It is possible to alter collagen to fabricate materials of various durabilities, structures and forms due to its adaptable nature. Collagen can also form complexes with different biologically active and medical substances [7]. Some of the applications of collagen are the formation of microspheres and microneedles for drug delivery [8], formulation of NPs for gene delivery [9], development of pellets and tablets for protein delivery [10], formation of gels and combination with liposomes for sustained drug delivery [11], cancer treatment [12] and collagen shields in ophthalmology [13]. The collagen types that can be purified and isolated for usage in pharmaceutical industries for drug delivery are: (i) Enzyme and alkali treated collagen, (ii) natural salt-soluble collagen,

(iii) insoluble collagen and (iv) acid-soluble collagen [14]. The degradation of collagen-based biomaterials for TE applications could possibly lead to tissue functionality and structure restoration. A few applications of collagen in TE have been utilised in cornea and skin related treatments [15–17], osteochondral defects [18], wound dressing, dermal filler and delivery systems [17]. Collagen-based wound dressings are used for applications of wound and burn coverage, ulcer treatment, therapeutic enzyme immobilization, antithrombogenic surfaces and bone filling substances [3]. When combined with elastic which provides flexibility, collagen supports and provides firmness and strength to body tissues and organs [18]. The usage of collagen as skin substitutes led to its usage in bioengineering of tissues such as ligament and blood vessels [19].

Particles with diameters 1–100 nm are called NPs. DDS based on NPs display enhanced efficacy of drugs and improve the drug's half-life, hydrophobic drug solubility and controlled and sustained drug release in the infected regions. Stimuli-responsive NPs regulate drug biodistribution and reduce drug toxicity [20,21]. TE requires the local controlled delivery of these bioactive and contrast compounds to deploy control over and monitor the tissues that are engineered. This need of TE is satisfied by NPs as they control scaffold characteristics such as mechanical strength and yield regulated release of bioactive compounds [22,23]. NPs overcome limitations such as unstable bioactivity, reduced half-life of bioactive and contrast compounds and low solubility, which makes them favourable for bioactive compound delivery and monitoring applications such as specific site imaging, cell patterning, biosensing, DNA structure probing and detection of molecules [23–25].

Polymeric NPs possess reduced cytotoxicity, high permeation and retention (EPR) effect, good biocompatibility and are able to deliver drugs which are poorly soluble and also control their release. They retain bioactivity of bioactive compounds from degradation of enzymes which makes them suitable to overcome problems in tissue engineering and regenerative medicine (TERM) applications [20,26]. Current PNPs systems are sensitive to stimuli such as temperature, light, pH, oxidizing/reducing agents, magnetic fields and enzymes which increases efficiency and specificity for TERM applications [21] king of collagen with NPs results in stabilization of the nanoparticles and helps with entrapment of the drug, to attain steady and regulated drug release for ideal therapeutic reactions. Collagen associates and absorbs NPs, leading to significant impacts on their biological functioning in any biofluid [27,28]. Collagen NPs are advantageous over other natural and synthetic polymeric NPs as they have favourable biocompatibility and biodegradability, low antigenicity, high contact surface, reduced toxicity and high cationic-charge density potential as they possess many amino groups; in the absence of surface modification with target compounds, they have a restricted capacity to cross the blood brain barrier. They are small sized, have large surface area and absorptive capability and the ability to diffuse in water to form colloidal solutions. Collagen NPs are also sterilized easily, thermally stable, improve cell retention and decrease effects of toxic by-products formed during degradation [29]. This is supported by experiments conducted by Alarcon et al. in which silver NPs when formulated with collagen type I showed great stability. In this study, spherical silver NPs with a diameter of 3.5 nm, stabilized in collagen, were prepared by a photochemical method at room temperature. This nanocomposite (AgNP@collagen) based on collagen displayed a nontoxic nature and remarkable biocompatibility and antibacterial characteristics. When examined, it was proved that bactericidal characteristics shown by AgNP@collagen were due to the presence of NPs [30].

This review will discuss the examples of collagen enhancing the properties of NPs, their chemical and physical preparatory methods and various developments and applications of these collagen NPs in DDS and TE.

## 2. Methods of Collagen Nanoparticle Preparation

Protein nanoparticle preparation and the process of drug encapsulation are carried out in moderate conditions, avoiding toxic organic solvents or chemicals. Collagen NPs

can be fabricated using three methods: (i) chemical, (ii) physical and (iii) self assembly. The chemical methods are coacervation or colyelectrolyte complexation and emulsification; physical methods include nano spray drying and electrospraying; and self-assembly comprises desolvation [31]. Some additional methods which have been discussed are milling, interfacial polymerisation, phase separation and polymer chain collapse [10,32]. This section discusses these collagen nanoparticle preparation methods which have specific advantages and limitations.

## 2.1. Emulsification/Solvent Extraction

The drug is dissolved in an organic protein or polymer solution and an emulsion system is formed by mechanical agitation. The polymer and drug are precipitated in droplets and NPs are formed by extracting the solvent by evaporation [33,34]. The double emulsion method has been used currently for greater efficiency of encapsulation [35]. In this method, a surfactant is used in dispersion for emulsion stabilization. The organic solvent is then taken out to conserve the NPs in an aqueous buffer. The double emulsion method of nanoparticle preparation is quick and economical. A stabilizing agent and surfactant are required to ensure the emulsion is stable as it is normally unstable thermodynamically. They affect interactions between the drug and matrix along with the drug release rate [31]. More specifically, at room temperature, an aqueous phase of collagen along with water and a hydrophilic surfactant, an organic phase consisting of a surfactant of lipophilic nature and a solvent miscible in water are mechanically agitated, forming a homogeneous emulsion system. This emulsion combined drop by drop with hot oil results in collagen nanoparticle fabrication [36]. An experiment conducted by Sharkawy et al. was to develop new stable Pickering emulsions that are stabilized with chitosan/collagen peptide NPs for possible cosmetic applications. Their study aims to understand the stability, microstructure and rheological properties of the Pickering formulations produced. Since both the polymers utilized in nanoparticle production have skin benefits, the study investigates the fate of the NPs after skin application by tracking their skin distribution following the penetration of the emulsion droplets. During the complex chitosan/collagen peptides nanoparticle formation, the long chitosan chains fold around the collagen peptides and form a new phase complex having different properties from that of the individual components [37]. The process of Emulsification/Solvent Extraction has been illustrated in Figure 1.

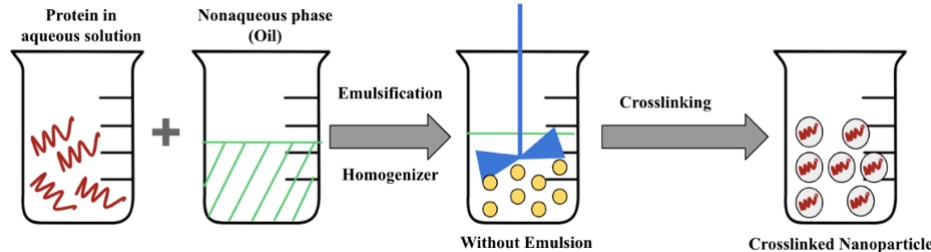

**Figure 1.** Emulsification/Solvent Extraction—The drug is dissolved in the polymer solution and an emulsion is formed by mechanical agitation. The polymer and drug are precipitated in droplets and NPs are fabricated by solvent extraction by evaporation.

## 2.2. Complex Coacervation/Polyelectrolyte Complexation

Complex coacervation or polyelectrolyte complexation is a process where oppositely charged polyelectrolytes in an aqueous solution undergo a liquid−liquid phase separation (LLPS) for the formation of a polymer-dilute and polymer-dense (coacervate) phase. The coacervate's properties can be adjusted by the chirality of the polyelectrolytes, salt concentration, pH, charge density, temperature and ionic strength, enabling the understanding of LLPS from changes in the physical environment [38]. It is a chemical nanoparticle preparation method and involves the incorporation of natural salt or alcohol to a solution of collagen which changes the collagen structure, to which a crosslinking material is added, leading to nanoparticle formation, as explained in Figure 2. This preparation method is

comparable to the desolvation method, only differing in parameters such as pH, rate of addition of solvent, temperature, homogenizer speed, crosslinking agent concentration, molar ratios of protein and organic solvent [36]. Due to the amphoteric nature of proteins which consist of multiple charged functional groups, they can be changed to either anionic or cationic just by modifying parameters such as pH. The protein which is charged can associate electrostatically with different polymeric electrolytes [31]. This method is useful in applications such as entrapment of DNA for gene therapy [39]. A study conducted by Singh et al. explores the potential of DNA NPs prepared with native collagen (NC) and methylated collagen (MC) to efficiently deliver genes into cells. The transfection abilities and physicochemical properties of these two types of NPs were studied in parallel. NC/DNA and MC/DNA NPs were prepared using the complex coacervation method. It was observed that NC formed complexes with the DNA at a low pH value. This complex accumulated promptly at neutral pH and did not give out significant protection to DNA because of its poor stability in serum. MC carries a positive charge at neutral pH and thus has higher stability under physiological conditions and strong DNA binding ability. It also showed that MC/DNA NPs were smaller and more stable than NC/DNA particles, which released DNA in a prolonged manner for up to 3 weeks [38,40].

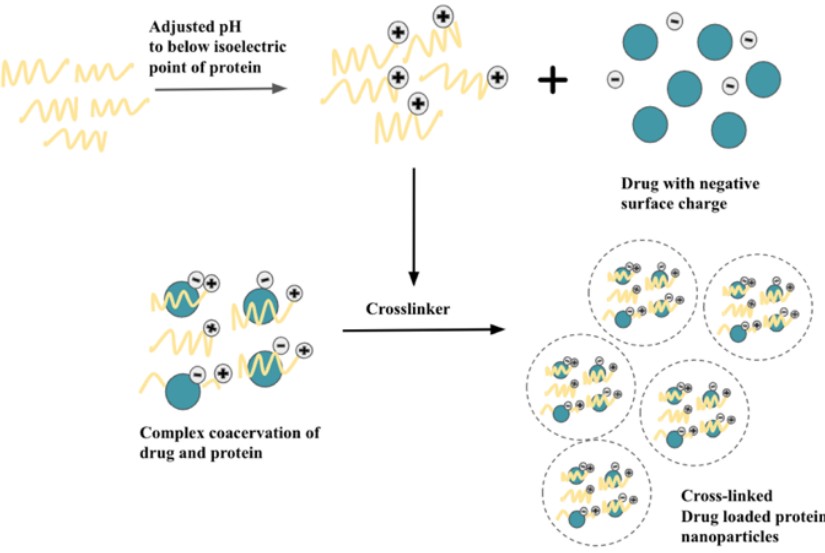

**Figure 2.** Complex Coacervation—This method regulates the pH to produce the protein cation or anion, and then reacts with different polymers to fabricate NPs. It involves the incorporation of natural salt or alcohol to a protein solution which changes the protein structure, to which a crosslinking material is added leading to cross-linked drug loaded protein NPs.

### 2.3. Phase Separation

Aqueous and organic phase separations are the backbone of the emulsion solvent evaporation technique of fabricating NPs. The polymers are put down in an organic solvent and the addition of a surfactant to this aqueous solution takes place, to avoid the emulsion particle fusion [41]. This solution is further put through ultrasonication, a method for mixing. Consequently, the droplets of the polymer are formed and the solvent is isolated. This is often concluded by the organic phase evaporation. The leftover solution consists of the polymeric NPs that are accumulated with the help of a centrifuge. Phase separation fabricates particles in the size range of 50–500 nm, which is modulated by changing the polymer solution concentration [42]. This method is well-known because of the conjugated polymer availability which is demonstrated in the experiments carried out by Yoon et al. where they prepared NPs of conjugated polymers from phase separation, which resulted in lipid-assembled NPs which can be combined with many functionalities such as inorganic/organic nanomaterials, cell specific ligands polyethylene glycols for in vivo applications [43]. The method of coacervation also involves phase separation,

which needs the liquid solutions' separation, one that contains the solvent and the other is a protein polymer. Coacervation is caused by the disruption of equilibrium through means such as salt addition. The charges create electrostatic forces which induce the formation of NPs. This process reduces the protein solubility which leads to phase separation [44] as shown in Figure 3.

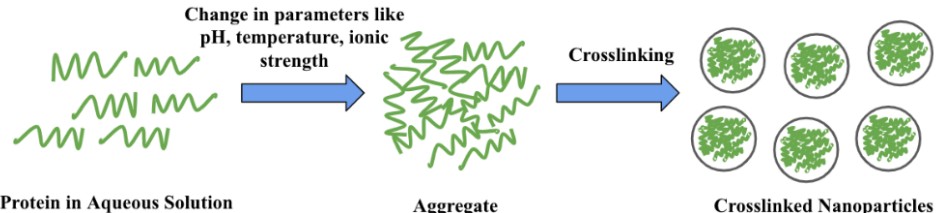

**Figure 3.** Phase Separation—Polymers are placed in an organic solvent and the addition of a surfactant to this aqueous solution takes place, to avoid the emulsion particle fusion. This solution is further put through ultrasonication, a method for mixing. Polymer droplets are formed and the solvent is isolated, concluded by the organic phase evaporation.

*2.4. Nano Spray Drying*

Nano spray drying is a physical nanoparticle preparation method which is used for spherical collagen nanoparticle fabrication in liquids. A dilute collagen solution is sprayed into chambers at an increased temperature where hot carbon dioxide and nitrogen gas flows in the direction of the spray from the nozzle producing hollow nanospheres which are collected by an electrode at the bottom, as demonstrated in Figure 4 [45]. Liquid nitrogen is used when collagen solution is sprayed to prevent denaturation which may be caused by high temperatures. Prepared collagen NPs are then frozen, their hardness is improved by tempering and they are lyophilized, crosslinked with specific agents and sanitized [10]. This is a rapid and inexpensive method to produce small-scale collagen NPs. Hydrophilic drugs can be encapsulated in spray dried NPs [46]. Spray drying is advantageous as the particle size can be regulated by varying parameters, such as nozzle size and spray rate. For protein NPs, surfactants are added to ensure stabilization of particles of the polymer [31,45].

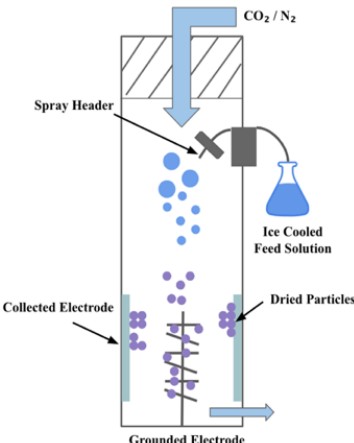

**Figure 4.** Nano Spray Drying—A liquid jet stream of the protein is released into a chamber using a nozzle and dried, and heated carbon dioxide and nitrogen and hollow nanospheres are collected by an electrode at the bottom of the chamber.

*2.5. Electrospraying*

As shown in Figure 5, a high voltage is applied to a solution of protein and the nozzle sprays a liquid jet stream resulting in the formation of an aerosolized droplet which consists of protein NPs in the colloidal size [47,48]. Drugs are easily able to be incorporated

into these NPs using this process [49]. Depending on the DDS type, parameters such as voltage applied, operating distance, gauge diameter of the needle and flow rate vary. An increased voltage is applied to the solution of the polymer to make sure it comes out of the syringe as NPs [50]. The electrospraying method is economical, easy to carry out and has good efficiency of encapsulation. Stable NPs can be fabricated without problems regarding biocompatibility and lowered encapsulation efficiency. The newer method of coaxial electrospraying consists of a coaxial spray head so both solutions can be guided to the electric field [31]. In experiments conducted by Nagarajan et al, solid collagen NPs were fabricated in a single step under ambient pressure and temperature conditions by electrospray deposition. Electrospraying of the collagen solution, increasing the solution conductivity then using salts to induce structural perturbation of the collagen molecules formed solid NPs. These solid collagen particles had high potential to act as drug carriers and this was shown by utilizing theophylline as a model drug using the coaxial spray technique. Release of theophylline was regulated by crosslinking of the molecules of collagen. This proved that electrospray deposition was a favourable preparatory method for fabrication of solid collagen NPs to be used in drug delivery applications [51].

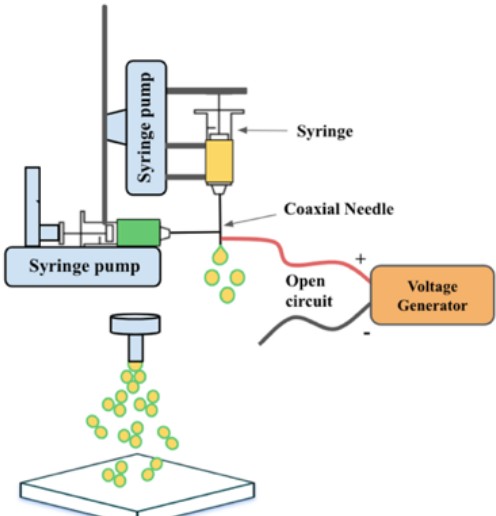

**Figure 5.** Electrospraying—NPs are fabricated by releasing a liquid jet stream through a nozzle (coaxial needle), forming an aerosolized droplet after exerting high voltage to the protein solution. High voltage is applied to the polymer solution to make sure it comes out of the syringe as NPs.

### 2.6. Self Assembly

Proteins that are modified hydrophobically, when added to aqueous solutions, can be self-assembled to form micelle NPs. These hydrophobic cores can be a channel for active molecules. In this method of self-assembly, chains of individual proteins are dissolved in solution which exceeds the critical micelle concentration, at a critical solution temperature to enable the formation of nanoscale aggregates [52]. Upon the formation of a bridge between the chains, these micelles can be made steady through the process of solidification. A number of studies have reported that collagen-based wound dressings engineered via self-assembly have been demonstrated to promote fibroblast production and accelerate wound healing [53]. The controlled release of biomolecules in collagen-based bioactive wound dressings can be achieved by varying the self-assembly conditions of the collagen constructs [27,54]. Vedanayagam et al. prepared varied sizes of silver NPs (AgNPs)—10 nm, 35 nm, and 55 nm—using nutraceuticals such as Pectin as stabilization and reducing agents through the method of microwave irradiation. These AgPectin NPs were accumulated in the self-assembly process of collagen which led to fabricated Collagen-Ag-Pectin nanoparticle-based scaffolds. The in-vitro biocompatibility analysis revealed that the collagen-Ag-Pectin NP-based scaffold showed greater antibacterial activity and

improved cell viability towards keratinocytes. Their study opened up the potential of utilizing the pectin caged silver NPs to develop collagen-based nanoconstructs for various biomedical applications such as drug delivery and TE.

### 2.7. Desolvation

Desolvation or simple coacervation is a self-assembly method which involves the incorporation of a desolvation factor such as a natural salt or alcohol to a collagen solution containing a drug. This desolvation factor changes collagen's structure and decreases its solubility. After a critical amount of desolvation, a crosslinking agent such as glutaraldehyde is added to the formed mass of collagen which results in nanoparticle formation [36]. It is widely used in the fabrication of protein NPs [55]. Protein-based NPs which are fabricated by desolvation can alter the size of particles relative to conditions such as concentration of the protein, pH, additive speed of the desolvating factor and temperature. Decreased concentration of the protein and increased pH produces NPs which are smaller in size. Concentration of the protein takes place by decreasing the protein solubility by addition of the desolvating factor [31]. This process has been illustrated in Figure 6.

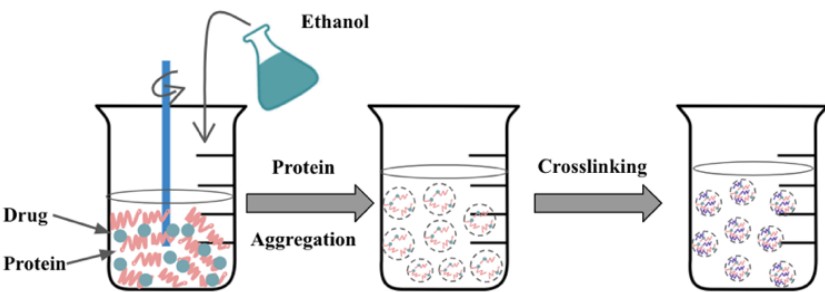

**Figure 6.** Self-assembly—Desolvation—Individual protein chains are dissolved in a solution at a Critical Solution Temperature which exceeds the Critical Micelle Concentration and leads to the generation of protein micelles during the nanosized aggregate formation, in the self-assembly method. In desolvation, NPs are fabricated by the addition of a desolvating agent to a protein solution which contains drugs.

### 2.8. Milling

Collagen in the nano-scale can be produced through the process of milling. In this process, a polymer material is disintegrated into finer NPs by the application of mechanical energy. This method uses milling balls for carrying out the high energy mechanical collision for the polymer disintegration (Figure 7). It is an economical method to carry out the reduction of the size of the particle on a large scale of production [56]. Heat energy in the process is produced as a result of the kinetic and mechanical energy that exists in the vessel used for milling [57]. Due to this production of heat energy, the milling vessel has to be cooled in order to avoid degradation of the material or overheating. Collagen, being a material sensitive to temperature, is milled mechanically utilising liquid nitrogen to avoid denaturation from heat, at cryogenic temperatures (below approximately −150 degree Celsius) [32]. Experiments were conducted to fabricate small-sized collagen NPs (average size: 200 nm) using the milling technique. Micron-scale collagen powder was collected and added into liquid nitrogen for half an hour. The cryo collagen was put in vacuum for 2 days to sublimate water from collagen. Freeze-dried collagen was then subjected to high energy milling to prepare very fine nano-scale collagen particles. For every milling round, the amount of freeze-dried collagen was 1:5 by weight ratio with milling balls in a milling container (120 mL volume). These collagen NPs have been looked into using cell culture tests with osteoblasts, and resulted in 30% enhancement of growth of the cell when compared to collagen particles in the micro scale [58].

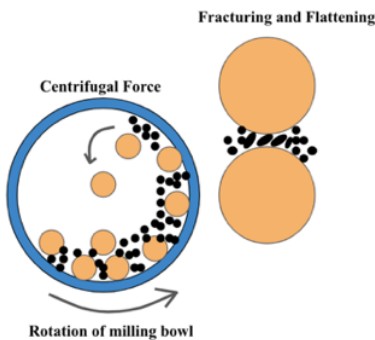

**Figure 7.** Milling—A polymer material is broken down into finer NPs by the application of mechanical energy by the rotation of a milling bowl. There are milling balls for performing high energy mechanical collisions for the polymer breakdown, using centrifugal force.

## 2.9. Interfacial Polymerization

Interfacial polymerization is a method used to produce polymer membranes and particles. It is a polymerisation technique which takes place between the two immiscible phase interfaces and results in the formation of a polymer which is strained to the interface [59]. In this process, an interfacial polymer is developed at the protein expression site following the polymer straining with a suitable dye. It provides these polymer materials with distinctive chemical and topological characteristics such as alternative surface chemistry, hollow structures or anisotropic shapes. The materials synthesized can be classified into zero-dimensional NPs, 2D films, 3D composite membranes and 1D nanofibers [60]. This process is explained in Figure 8.

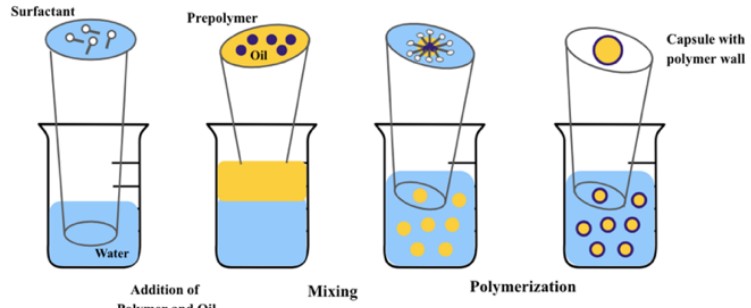

**Figure 8.** Interfacial polymerization—It occurs between the two immiscible phase interfaces, and results in formation of a polymer strained to the interface. An interfacial polymer is developed at the protein expression site following the polymer straining with a suitable dye. The figure shows formation of a prepolymer upon the addition of polymer and oil, and further mixing and polymerisation leads to the formation of a capsule with polymer wall.

## 2.10. Polymer Chain Collapse

Single-chain polymer NPs (SCNP) are fabricated using the polymer chain collapse method, which can fabricate particles with great stability in the 5–20 nm range [61]. Distinct molecules are produced due to the control of the precursor chain which can direct the nanoparticle properties [62]. Different kinds of the SCNP method exist and the reaction type is based on the functional groups that are involved. Intramolecular crosslinking is, however, more advantageous than the intermolecular cross-linking for these methods [63]. In homofunctional polymer chain collapse, a functional group which can attach with itself on the precursor chain is placed, after which a reaction that couples the functional group is performed (single-chain polymer NPs). However, particles which are not spherical in shape are produced using this method. Heterofunctional coupling, which needs two perpendicularly cross-linked functional groups, has therefore been looked into for better

results. It is shown through data that this method can produce NPs with a better globular shape [44,64]. Intramolecular crosslinking of a cross linkable polymer is displayed in Figure 9.

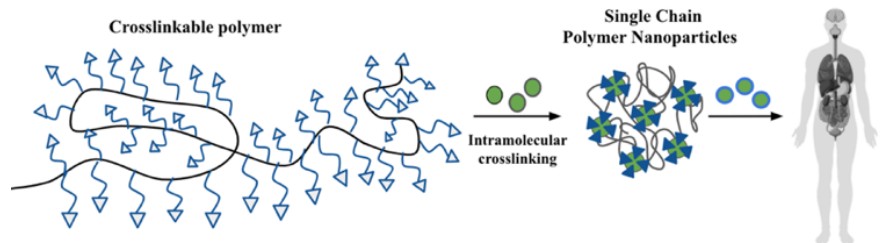

**Figure 9.** Polymer Chain Collapse—Distinct molecules are fabricated due to the precursor chain control that can direct the nanoparticle properties. Through intramolecular crosslinking, the polymer is fabricated to form single chain polymer NPs that find various applications in the human body.

The advantages and limitations of these various methods of collagen NP preparation have been discussed in Table 1.

**Table 1.** Advantages and limitations of collagen NP preparation methods.

| Preparation Method | Advantages | Limitations | References |
|---|---|---|---|
| Emulsification/Solvent Extraction | Process is simple, equipment required is simple, recovery can be controlled, high flexibility and selectivity | Require stabilizer and surfactant because of unstable thermodynamic nature, need to add organic solvent and then remove it, residues of organic solvent may be toxic | [10,34] |
| Complex Coacervation/Polyelectrolyte Complexation | Particles formed are very stable, NPs of smaller size, by guiding process conditions, nanoparticle size and shape can be controlled, can be combined with sensitive drugs | Hard process to scale up | [36,65] |
| Phase Separation | Specialized apparatus is not necessary, particle size controllable by altering polymer solution concentration, uniform particles are formed | Limited particle size diameter, small-scale production, organic solvent requirement | [43,44] |
| Nano Spray drying | Economical process, simple to carry out experimentally, encapsulation of hydrophilic drugs takes place easily, beneficial for heat-sensitive samples as it helps maintain temperature of the nanoparticle droplets | Small scale production, difficult to integrate hydrophobic drugs, reduced encapsulation efficiency, great energy consumption | [45,46] |
| Electrospraying | Can be scaled for industrial use, good drug loading efficiency, ease of particle synthesis due to single-step processing, formation of dry particles | Reduced flow, could produce degradation of macromolecules | [49,50] |
| Self-Assembly | Highly stable process, small NPs can be formed with high encapsulation efficiency | Hard to control NP size, shape and the potential of protein strain exists | [27,54] |
| Desolvation/Simple Coacervation | Increased encapsulation efficiency, Size and shape of NPs can be controlled using reaction conditions. | Can be carried out only for proteins influenced by dissolution or diluted by carrier proteins | [31,55] |
| Milling | Economical, easy experimentation, controllable NP size, large scale prodcution | Chamber has to be cooled due to heat release, uncontrollable NP shape, can be carried out only for coarse NPs | [57] |
| Interfacial Polymerization | Easy to carry out, inessential monomer purity | Expensive polymer monomer, takes a lot of time to be carried out | [59,60] |
| Polymer Chain Collapse | Controllable NP properties, high stability, enhanced spherical shape particle production | Limited particle diameter, hard to control side reaction occurrence | [62,63] |

## 3. Collagen-Based NPs in Biomedicine

Collagen NPs are good prospects for regulated drug release strategies as their physical characteristics such as absorption capacity, surface area and size are simple to configure [44,66]. They are also suitable for cell and gene delivery systems as they can defend the environment of the body [67]. Temperature, pH and molecular weight of collagen are determining factors for the stability of collagen NPs. Dermal delivery of retinol present in collagen NPs showed that retinol was more stable due to enhanced properties of these collagen NPs (Figure 10) and it displayed a faster transportation of incorporated drugs through the skin.

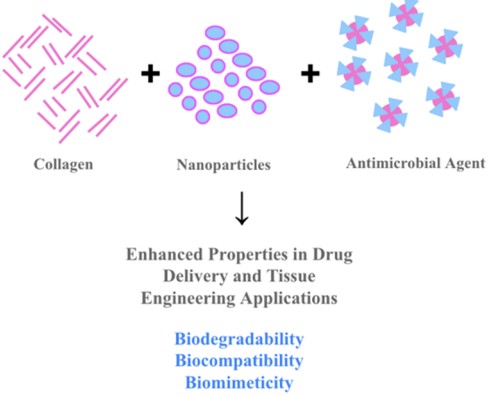

**Figure 10.** Enhanced properties of collagen NPs in DDS and TE.

The formation of NPs is guided by electrostatic forces along with dissolving reagents for higher inter-charge reactions between collagen and plasmid DNA. They can be absorbed by the reticuloendothelial system and allow the amplified uptake of drugs into macrophages and other cells of the body [68]. This ability of the collagen NPs enables them to act as systemic delivery carriers for therapeutic compounds and cytotoxic agents [3]. Collagen nanoparticle scaffolds can effortlessly enter wounds in the skin, which makes them ideal for wound healing and regulated delivery of drugs [29]. Other applications of collagen based NPs in the biomedical field include vascular, bone and skin grafting, insertion of bio scaffolds, wound healing fillers, cartilage and nerve tissue regeneration [32]. The following section will discuss some of the notable experimental applications of collagen NPs in drug delivery and TE.

### 3.1. Collagen Based NPs in DDS

Rathore et al. aimed at investigating the neuroprotective effect of the silymarin—collagen nanoparticle DDS. They studied the importance of NPs in improving the therapeutic effect of silymarin against neuronal injury. Collagen NPs were fabricated and stabilized using malondialdehyde(MDA) and 3-ethyl carbodiimide-hydrochloride (EDC-Hcl) as crosslinking agents. The collagen NPs that were fabricated possessed a loading efficiency of $3.17 \pm 0.37\%$, entrapment efficiency of $76.7 \pm 2.4\%$, and displayed a regulated and slow release. Their study concluded that the encapsulation of silymarin into collagen NPs improved the therapeutic efficacy of silymarin by enhancing its bioavailability [69]. Surface modified collagen nanospheres which are delivered to the bloodstream are another favorable method for drug delivery to the brain. Modification of the surface in the spheres enables the identification of the particular surface receptors of the cell which allow their transcytosis through modification of the sphere properties such as surface charge and hydrophobicity, or blood-brain barrier (BBB) [70,71]. Wohlfart et al. studied that the modification of surface in these nanospheres may allow surface receptor recognition of a particular cell which, in turn, allows the modification of the physiological properties of the nanospheres or the transcytosis through BBB, to help them pass over BBB through adsorption carried out by endothelial cells [72–74].

Mondal et al. loaded gold NPs (Au) on hydroxyapatite (HAp) surface using a rapid microwave assisted technique. They prepared varied concentrations of Au-loaded nanostructures covered with collagen (Au–HAp–Col), which were enhanced for the loading and releasing of the doxorubicin drug, for applications in biomedicine. Collagen and Au-HAp NPs have an electrostatic interaction between them which makes a steady nanostructure. A ph-receptive release of approximately 53% and the highest efficiency of drug loading of approximately 58.22% were procured for 0.1 weight percent of Au-HAp-Col NPs. Their results demonstrated that these enhanced Au–HAp–Col NPs with bioactive and biocompatible characteristics could be useful for drug delivery, scaffold materials, cell growth, proliferation and adhesion [73]. In this study conducted by Suresh et al., silver NPs (DdAgNPs) were synthesized with collagen and doxycycline (DO) and its strength to be utilised as a bactericidal agent was evaluated. It was found that the DdAgNPs combined with collagen displayed stronger antibacterial action against all the test organisms compared to the DdAgNPs alone [74].

Another study showed that collagen NPs can be helpful in tumor infiltration for anti-cancer drug delivery. This is due to the advantage of collagen which resembles the microenvironment of the tumor. In this study, they developed collagen-based tumor spheroids and optimized them using 95-D, U87 and HCT116 cells. The antitumor and delivery efficiencies of the drug-conjugated NPs in this model were studied through cytotoxicity and uptake studies. Their results demonstrated that the conjugated NPs reach the tumor cells by perforating into the gel matrix. This model was found to be more precise in figuring out the therapeutic results and dynamics of the drug transport agents in vivo, and in the demonstration of tumor biology, which fastens the drug discovery for cancer therapy [75]. Studies have also demonstrated that collagen NPs have been utilised as parenteral carriers for therapeutic and cytotoxic substances, such as hydrocortisone (Berthold et al.) and camptothecin (Yang et al.) [8,76,77] These investigations on collagen in DDS are summarized in Table 2.

**Table 2.** Investigations on collagen NPs in DDS.

| Agent (Crosslinking/ Stabilising/Optimising) | Effect of Collagen NPs | Application | Reference |
|---|---|---|---|
| Malondialdehyde(MDA); 3-ethyl carbodiimide-hydrochloride (EDC-Hcl) | Enhanced bioavailability and improved therapeutic effect of silymarin drug | Drug for Neuronal injury | [69] |
| | Enables identification of the particular cell surface receptors which allows transcytosis | Drug delivery to the brain | [70] |
| | Enhancement of motor functions in PD model and cognitive functions in AD model | Nerve Growth Factor (NGF) delivered to the brain | [73] |
| Gold NPs (Au) on hydroxyapatite (HAp) surface | Enhanced Au–HAp–Col NPs with bioactive and biocompatible characteristics for loading and releasing of the doxorubicin drug | Drug delivery, scaffold materials, cell growth, proliferation and adhesion | [74] |
| Silver NPs (DdAgNPs) with doxycycline (DO) | Stronger antibacterial action against all the test organisms compared to the DdAgNPs alone | Bactericidal agent | [75] |
| 95-D, U87 and HCT116 cells | More precise therapeutic results and dynamics of the drug transport agents in vivo | Tumor infiltration for anti-cancer drug delivery | [77] |

### 3.2. Collagen Based NPs in TE

Gold NPs were incorporated into collagen nanoparticle scaffolds which later on reacted with growth factors and molecules for cell adhesion. This helped in reduction of inflammation and formation of granulation tissue with no problems of rejection, which make them perfect for wound healing [32]. Volkova et al. conducted experiments which displayed that gold NPs along with cryopreserved human fibroblasts administered top-

ically to burns and wounds increased the rate of healing and enhanced collagen deposition [32,78]. Vedhanayagam et al. fabricated scaffolds based on silver, pectin and collagen NPs which supplied greater antibacterial activity and improved viability of the cell toward keratinocytes [79]. Patrascu et al. prepared scaffolds of silver and collagen NPs for burn and wound healing, and skin repair. Hydroxyapatite silver collagen nanoparticle composite scaffolds acted as prospective bone graft materials [80,81]. Nidhin et al. developed a construct of collagen NPs for TE applications and imaging by crosslinking $\alpha$-$Fe_2O_3$ NPs capped with starch to collagen. This construct had enhanced mechanical properties, had better crosslinking, was able to provide better viability to the cell, had greater super paramagnetic behaviour and could be used in imaging and as bio implants [82,83].

Moon du et al. prepared tissue constructs based on skeletal muscles with acellular tissue scaffolds based on collagen NPs which enhanced contractile force generation [80,84]. Wang et al. studied the production of collagen nanoparticle bone and development for preservation of alveolar ridge. For protection of the residual ridge after extraction of the tooth, artificial collagen nanoparticle bone was introduced into the patient. Scans taken showed that the residual ridge had combined with the collagen nanoparticle bone which also had a greater alveolar bone mineral density [35,85]. Shen et al., after trials on rabbits, displayed enhanced rate of solid fusion and bone mineral density when bone based on collagen NPs along with stem cells derived from adipose and allogene was utilized. They could also be used to repair defects in the human ulna [86]. Quinlan et al. prepared porous scaffolds using collagen glycosaminoglycan (CG) and bioactive glass to promote TE in the bone [87,88]. Collagen and hydroxyapatite could be manufactured to form a composite of collagen-hydroxyapatite by lyophilization and dehydrothermal treatment [89,90]. For implants in vascular medicine, collagen NPs are used as a scaffold in tissue engineered vascular grafts with the addition of different compounds. Collagen NPs behave as a template for growth into the vascular graft and cell recognition [91].

Injections based on collagen NPs can be provided to locally deliver therapeutic components or drugs to hinder neurodegeneration and they contribute structural foundation [70]. Zhang et al. showed the usage of collagen nanoparticle or nano-sized $\beta$-tricalcium phosphate conduits along with filaments of collagen and nerve growth agents for regeneration of facial nerves [90]. Amiri et al. conducted an experiment to study the effects of cell attachment of collagen NPs on crosslinked electrospun nanofibers. Collagen NPs enhanced various properties such as viability of the cell, adhesion and more spreading on the scaffold. Regarding crosslinking with glutaraldehyde, collagen NPs mimicked the extracellular matrix better than collagen nanofibers [91]. Renal cartilage sponge collagen NPs were utilized as an osmotic accelerator for hormone replacement therapy for transdermal delivery of 17$\beta$-estradiol-hemihydrate. Conclusions displayed that the hydrogels in which estradiol collagen NPs were present prolonged the release time of estradiol and improved the absorption of estradiol to a great extent. Hence, sponge collagen NPs can act as prospective carriers for transdermal drug delivery [92,93]. These investigations on collagen in TE are summarized in Table 3.

**Table 3.** Investigations on Collagen NPs in TE.

| Agent (Crosslinking/ Stabilising/Optimising) | Effect of Collagen NPs | Application | Reference |
|---|---|---|---|
| Gold NPs | Reduction of inflammation and formation of granulation tissue with no problems of rejection | Wound healing | [31] |
| Gold NPs and cryopreserved human fibroblasts | Increased rate of healing and enhanced collagen deposition | Burn and wound healing | [80] |
| Silver and pectin | Greater antibacterial activity and improved viability of the cell toward keratinocytes. | Dermal TE | [81] |
| Hydroxyapatite and silver | | Burn and wound healing, skin repair, bone graft materials | [82] |

**Table 3.** *Cont.*

| Agent (Crosslinking/Stabilising/Optimising) | Effect of Collagen NPs | Application | Reference |
|---|---|---|---|
| α-Fe$_2$O$_3$ NPs capped with starch | Enhanced mechanical properties, better crosslinking, better viability to the cell, greater super paramagnetic behaviour | Imaging, bio implants | [83] |
| | Enhanced contractile force generation | Tissue constructs based on skeletal muscles with acellular tissue scaffolds | [84] |
| | Greater alveolar bone mineral density | Protection of residual ridge after tooth extraction, artificial collagen nanoparticle bone. | [85] |
| Stem cells derived from adipose and allogene | Enhanced rate of solid fusion and bone mineral density | Repairing defects in the human ulna | [86] |
| Collagen glycosaminoglycan (CG) and bioactive glass | | Bone TE | [87] |
| | Acted as template for growth into the vascular graft and cell recognition | For implants in vascular medicine, scaffold in Tissue Engineered Vascular Grafts | |
| Nano-sized β-tricalcium phosphate and nerve growth agents | | Regeneration of facial nerves | [92] |
| Glutaraldehyde | Enhance cell viability, adhesion and more spreading on the scaffold | Mimicking extracellular matrix (ECM) | [93] |
| | Prolonged release time of estradiol and improved absorption of estradiol | Osmotic accelerator for hormone replacement therapy for transdermal delivery of 17β-estradiol-hemihydrate, carriers for transdermal drug delivery | |

## 4. Conclusions

Collagen is one of the most utilized biodegradable polymers as it provides ideal polymeric aid for DDS, while behaving like a natural biomaterial with wound healing and haemostatic properties [7]. Collagen-based biomaterials are extremely important for regenerative medicine and TE. Due to its decreased immunogenicity and exceptional biocompatibility, collagen is the selected protein for the fabrication of biomaterials [94]. Enhancement of mechanical strength, biodegradability and delivery characteristics support the optimization of biomaterials based on collagen for biomedical applications [95]. The experimental studies carried out by researchers, some of which have been reviewed in this paper, demonstrated that in vivo degradability, absorption and drug delivery are modulated by the physical or chemical crosslinking of collagen, to regulate the effect of drug delivery [96]. Collagen NPs or nanospheres are thermally stable particles which promptly attain their sterilization and act as an effective carrier for therapeutic compounds and cytotoxic agents [97]. Due to their reduced size, high capacity of adsorption, water dispersion ability to form clear colloidal solutions and high surface area, collagen NPs display high potency to be utilised as formulations for regulated release of steroids or antimicrobial agents [7]. Methods of nanoparticle preparation such as polyelectrolyte complexation and desolvation are widely used, whereas nano spray drying is up and coming. As mentioned in Table 1, all methods have their limitations and benefits and, therefore, there is potential for carrying out vast research to overcome these disadvantages. Drug release efficacy and drug loading of proteins are also adjusted based on nanoparticle characteristics or drug type and concentration. Efficiency of drug transfer can be increased

by controlling and maintaining features such as surface charge, shape and size. To produce ideal collagen NPs, the correct process and materials should be selected according to the relevant application [31]. In the coming years, smart collagen NPs will be able to engage and guide stem cells to specific body sites and dictate the development of in vivo tissues [98]. Collagen plays a crucial role in biomedicine and it is clear from the facts and discussion in this review that collagen NPs are up and coming materials in drug delivery, formation of tissues, proliferation and attachment of cells [34,99,100].

**Author Contributions:** Conceptualization, A.L. and A.A.; methodology, P.M.; software, A.A.; validation, A.L., P.M. and A.A.; formal analysis, A.A.; investigation, A.L.; resources, P.M.; data curation, A.A.; writing—original draft preparation, P.M. and A.A.; writing—review and editing, A.L. and S.R.; visualization, A.L.; supervision, A.L. and S.R.; project administration, H.L., S.R.; funding acquisition, S.R., H.L. All authors have read and agreed to the published version of the manuscript.

**Funding:** This research received no external funding.

**Institutional Review Board Statement:** Not applicable.

**Informed Consent Statement:** Not applicable.

**Data Availability Statement:** Data is contained within the article.

**Acknowledgments:** The authors would like to thank Manipal Institute of Technology and Manipal Academy of Higher Education for providing them with assistance and resources to help develop their research interests.

**Conflicts of Interest:** The authors declare no conflict of interest.

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
