# Peer review of "Collagen Nanoparticles in Drug Delivery Systems and Tissue Engineering"

_applsci, doi:10.3390/app112311369_

Round 1

Reviewer 1 Report

The focus of introduction should be on how "collagen" and "nanoparticle" synergize to facilitate biomedical applications. There is a lack of description on why collagen nanoparticle is better than other nanoparticles. 

Section 3 should be the most important section but it is very general and does not have any figures. 

Several claims in Table 1 and Section 2 are not accurate. For example, scale up synthesis using polyelectrolyte complexation should not be a major challenge. Disadvantages from different methods such as size control overlap significantly. There is a lack of perspectives or discussions on how different methods can compliment each other to make better nanoparticles.

Overall, major concerns remain in the significant lack of descriptions on the importance and applications of collagent nanoparticles. I cannot be convinced to recommend acceptance.

Author Response

Response to Reviewer 1:

Dear Sir/Ma’am,

Thank you for your valuable comments on our manuscript. As per your feedback, we have made changes to our manuscript.

1) Additional information about collagen nanoparticles and its importance over other nanoparticles has been explained in the main manuscript.

2) More information has been added to all sections including Section 3, along with a diagram highlighting a few significant enhanced properties of collagen nanoparticles combined with antibacterial agents, for usage in Drug Delivery and Tissue Engineering applications. This section has been divided further into two sub-sections for Drug Delivery and Tissue Engineering applications respectively, for better clarity, and tables have been added to summarise the studies.

We have made a sincere attempt at making this section less generic to justify it being the most important one.

3) The claims made in Table 1 and Section 2 have been crosschecked and verified. The advantages and limitations have been altered based on recent information, as per the suggestions made. Overlapping information has been checked.

We also studied how different preparation methods complement one another. We’ve elaborated on how Coacervation is related to Phase separation method. We have also added experimental information and practical works for Coacervation, Emulsification, Electrospraying, Milling and Self Assembly. Desolvation method is a self-assembly process and they have been explained in detail in the review.

We hope that the changes made to the manuscript in terms of descriptions, relevant information, and English language check are able to demonstrate potential for publication.

Thank you.

Reviewer 2 Report

The review manuscript by Ashni Arun et al., titled Collagen Nanoparticles in Drug Delivery Systems and Tissue Engineering, is fluently presenting the obtaining methods of collagen nanoparticles, as well as their application in TE.

Still, the review is not based on recent studies published in the last two years, so I strongly encourage the authors to update it.

The preparation methods are presented in a very didactic, and less scientific manner, therefore I recommend completing them with some practical studies which can underline the challenges and the influence of different mentioned factors.  

The part regarding the applications in TE is better described, but recent references are missing, so it needs to be updated to 2021 year.

Author Response

Response to Reviewer 2:

Dear Sir/Ma’am,

Thank you for your valuable feedback on our manuscript. Based on the comments, we have added more papers from the past few years which are relevant to our review.

The available experimental studies based on a few collagen nanoparticle preparation methods have been added as per your feedback.

We hope that the changes made to the manuscript in terms of descriptions, relevant information, and English language check are able to demonstrate potential for publication.

Thank you.

Reviewer 3 Report

Collagen Nanoparticles in Drug Delivery Systems and Tissue Engineering has been written and presented well however the information presented and references cited in this work are not sufficient. The authors are encouraged to enlist various nanoparticle-based drug delivery systems investigated so far by several researchers to improve drug delivery.  In addition, investigations on collagen in tissue engineering are also required to be presented in tabular form. 

Author Response

Response to Reviewer 3:

Dear Sir/Ma’am,

Thank you for your valuable feedback on our manuscript. As per your suggestion, we have added more relevant information and references to the review.

More detailed information has been added to the entire paper. This section has been divided further into two sub-sections for Drug Delivery and Tissue Engineering applications respectively. The investigations of collagen in Drug delivery and Tissue Engineering have also been tabulated respectively.

We hope that the changes made to the manuscript in terms of descriptions, relevant information, and English language check are able to demonstrate potential for publication.

Thank you.

Round 2

Reviewer 1 Report

Thanks to the authors for the changes. Most of my previous concerns have been addressed.

Reviewer 2 Report

The authors updated the manuscript as required, therefore I agree with its publishing.

Reviewer 3 Report

None